# Mitigating Age-Related Ovarian Dysfunction with the Anti-Inflammatory Agent MIT-001

**DOI:** 10.3390/ijms242015158

**Published:** 2023-10-13

**Authors:** Min-Hee Kang, Yu Jin Kim, Min Jeong Cho, JuYi Jang, Yun Dong Koo, Soon Ha Kim, Jae Ho Lee

**Affiliations:** 1CHA Fertility Center Seoul Station, Seoul 04637, Republic of Korea; mhkang312@chamc.co.kr (M.-H.K.); yj_kim@chamc.co.kr (Y.J.K.); jminj725@chamc.co.kr (M.J.C.); 2Department of Biomedical Sciences, CHA University, Pocheon 11160, Republic of Korea; juij04370@gmail.com (J.J.); rndbsehd@gmail.com (Y.D.K.); 3Mitoimmune Co Ltd., Seoul 06253, Republic of Korea; shakim@mitoimmune.com

**Keywords:** aging, ovarian dysfunction, inflammation, ovarian restoration, RNA-seq, cytokines

## Abstract

Ovarian aging is a major obstacle in assisted reproductive medicine because it leads to ovarian dysfunction in women of advanced age. Currently, there are no effective treatments to cure age-related ovarian dysfunction. In this study, we investigated the effect of MIT-001 on the function of aged ovaries. Young and old mice were utilized in this study. MIT-001 was intraperitoneally administered, and the number of follicles and oocytes was analyzed. Each group was then retrieved for RNA and protein isolation. Total RNA was subjected to mRNA next-generation sequencing. Protein extracts from ovarian lysates were used to evaluate various cytokine levels in the ovaries. MIT-001 enhanced follicles and the number of oocytes were compared with non-treated old mice. MIT-001 downregulated immune response-related transcripts and cytokines in the ovaries of old mice. MIT-001 modulates the immune complex responsible for generating inflammatory signals and has the potential to restore the function of old ovaries and improve female fertility.

## 1. Introduction

Age-related female infertility is a natural process primarily associated with ovarian dysfunction, including a diminished ovarian reserve and a decline in oocyte quality [1,2]. This is a significant challenge in assisted reproductive technology (ART) treatments, particularly for women in their late 30s and 40s who may experience poor pregnancy outcomes [2,3,4]. Unfortunately, no treatments are available to restore the competence of oocytes and embryos in ART treatments and to overcome age-related ovarian dysfunction [2].

Ovarian aging is classified into two types: physiological aging and pathological aging [5]. In terms of physiological aging, advanced maternal age is a critical factor underlying ovarian dysfunction [6,7,8]. Recent research identified several cellular and molecular characteristics of aging, including telomere attrition, genome instability, autophagy decline, oxidative stress, and mitochondrial dysfunction [9,10]. Pathological ovarian aging is thought to be linked with immune reactions and inflammation and to cause a diminished ovarian reverse, primary ovarian insufficiency (POI), and a poor ovarian response [11]. However, the mechanisms underlying both types of ovarian aging are unclear [10]. Several anti-aging agents have been developed based on these characteristics to improve human health by preventing aging and rejuvenating cells [12,13]. Furthermore, several studies have used anti-aging agents to recover the function of old ovaries [10].

Inflammatory aging plays an important role in the pathogenesis of ovarian aging, suggesting that suppression of inflammation can prevent ovarian insufficiency [11]. Inflammatory factors such as interleukin (IL)-6 and IL-8 are likely expressed in the ovaries of individuals with POI. Many inflammatory markers have been identified, such as plasma tumor necrosis factor, ILs, and plasma inflammatory protein, which can be used to monitor ovarian function and overcome ovarian aging [11]. However, there are many other reasons for the occurrence of ovarian aging and therefore its pathogenesis is unclear [14]. There are several unanswered questions regarding the efficacy of anti-inflammatory treatments and other therapeutic approaches. For example, it is unclear if anti-inflammatory treatments have a more significant effect than other interventions or if individuals with genetic susceptibility to ovarian aging benefit from anti-inflammatory treatment. Additionally, it is unknown whether anti-inflammatory treatments can prevent ovarian aging. These questions require further investigation to improve understanding of the underlying mechanisms and potential interventions for the treatment of ovarian aging and infertility.

MIT-001, a cyclopentylamino carboxymethylthiazolylindole compound, serves as a novel anti-inflammatory agent with a broad spectrum of therapeutic effects against both inflammatory and degenerative diseases [15,16,17,18]. Notably, MIT-001 has demonstrated its ability to inhibit osteoclast differentiation in lipopolysaccharide-treated bone marrow cells of C57/BL6 mice, resulting in a significant reduction in degenerative bone loss [16]. Furthermore, NecroX-7, a subtype of MIT, has exhibited its capacity to mitigate acute graft-versus-host disease through the reciprocal regulation of type 1 helper T cells and regulatory T cells, along with the inhibition of key inflammatory mediators such as tumor necrosis factor (TNF), interleukin-6 (IL-6), and Toll-like receptor 4 (TLR4) [18]. Moreover, MIT-001 is a promising candidate to alleviate mitochondrial dysfunction upon oxidative stress due to its role in mitochondrial antioxidant defense [17]. Our group reported that MIT-001 restores homeostatic phenotypes of human placenta-derived mesenchymal stem cells upon TNF-α- and interferon-γ-induced inflammation [19].

Inflammatory processes are linked to mitochondria, and several factors including oxidative stress, cytokines, cell death signaling, and autophagy are involved in inflammation-related ovarian aging [11]. Mitochondrial dysfunction has been implicated in cellular senescence during ovarian aging [20]. The effects of anti-inflammatory agents on mitochondrial functional recovery in old ovaries and the underlying mechanism have not been elucidated. Treatment with an anti-inflammatory agent may lead to functional recovery of aged ovaries and improve the quality of oocytes and embryos. Anti-inflammatory agents may be utilized to facilitate clinical pregnancy in infertile women of advanced age. Elucidation of the mechanism underlying ovarian aging is important to overcome aging-related issues in ARTs, and the development of anti-inflammatory agents may provide a therapy. Aging increases inflammation, which can lead to a diminished ovarian reserve and a poor ovarian response. Therefore, recovery of ovarian function upon aging using anti-inflammatory agents is crucial.

This study aimed to assess the potential of the anti-inflammatory agent MIT-001 to mitigate ovarian dysfunction resulting from aging. We also investigated its effect on inflammatory signaling in ovaries by analyzing changes to the transcriptome via RNA sequencing (RNA-seq) and the assessment of cytokine.

## 2. Results

### 2.1. MIT-001 Treatment Enhances Follicular Development in Ovaries of Old Mice

We assessed the effects of MIT-001 on ovarian function in vivo. Young (positive control), old (negative control), and MIT-001-treated old (experimental group) female BDF1 mice were utilized according to the schematic diagram depicted in Figure 1A. Following MIT-001 injection, the body weight of old mice slightly and non-significantly decreased (Figure 1B). Hematoxylin and eosin (H&E)-stained ovarian sections demonstrated that the number of total follicles, including preantral (primary and secondary follicles) and antral follicles, was markedly higher in MIT-001-treated old mice than in old mice (Figure 1C). To evaluate the ability of MIT-001 to enhance follicular development, germinal vesicle (GV) oocytes were collected from the ovaries of each group of mice at 48 h after pregnant mare serum gonadotropin (PMSG) injection (Figure 1D,E). The numbers of primary, secondary, and antral follicles per ovarian section were analyzed (Figure 1D). The number of antral follicles per ovarian section was significantly increased in MIT-001-treated old mice compared with old mice. The numbers of GV oocytes retrieved from 12, 13, and 14 ovaries of young, old, and MIT-001-treated old mice, respectively, were evaluated. Although old mice exhibited a decline in follicular development compared with young mice, the average number of GV oocytes was significantly higher in MIT-001-treated old mice than in old mice (old = 3.50 ± 3.55 and MIT-001-treated = 5.53 + 1.91, *p* < 0.05), but was highest in young mice (9.14 ± 4.49, *p* < 0.005; Figure 1E).

### 2.2. Transcriptome Changes in Ovaries of MIT-001-Treated Old Mice

To investigate the effect of MIT-001 on gene expression, we performed RNA-seq using total RNA extracted from ovaries of young (8–9 weeks old, 2 ovaries), old (65 weeks old, 2 ovaries), and MIT-001-treated old (65 weeks old, 2 ovaries) mice. Appendix A show the gene targeting strategy, indel information, raw read data, and mapped gene counts processed using the pipeline presented in Figure 2A. Principal component analysis was conducted of differentially expressed genes (DEGs) among the three groups (Figure 2B). PC1 captured 55.16% (young vs. old) and 75.09% (young vs. treated) of the differences. PC2 showed that the difference between the treated and young groups (14.48%) was less than that between the treated and old groups (39.55%). The genes were analyzed using DESeq2 (Figure 2C and Appendix A) with the following three comparisons: (1) old vs. young, (2) MIT-001-treated old vs. old, and (3) MIT-001-treated old vs. young. In the old vs. young, treated vs. old, and treated vs. young comparisons, 829, 830, and 1077 genes were upregulated more than two-fold, respectively, and 385, 191, and 477 genes were downregulated more than two-fold, respectively. A Venn diagram showed that among 2732 DEGs screened in overall comparisons, 267 were differentially expressed in all comparisons (Appendix A). Gene set enrichment analysis using Gene Ontology (GO) datasets (C5) from the Molecular Signatures Database ([21], Figure 2D and Appendix A) revealed that a significant number of total DEGs were linked with biological process GO terms associated with the immune system, such as Mononuclear Cell Differentiation (gene count: *n* = 114; rich factor = 0.23), Positive Regulation of Cell Adhesion (*n* = 113, 0.24), Cytokine-mediated Signaling Pathway (*n* = 98, 0.25), Leukocyte Proliferation (*n* = 96, 0.27), Regulation of T-cell Activation (*n* = 91, 0.27), Regulation of Immune Effector Process (*n* = 118, 0.28), and Leukocyte Cell-Cell Adhesion (*n* = 108, 0.29).

We further defined four clusters in the heatmap of common DEGs based on differentially expressed patterns between groups (denoted by green, red, purple, and blue boxes; Figure 2E). Enrichment tests with DEGs in each cluster using GO and canonical pathway databases showed that the majority of DEGs (e.g., Cd300a, Cd74, Fgr, Fcer1g, C1qc, Fcgr4, and Tyrobp) related to immune responses (Immunity, Innate Immunity, Adaptive immunity, Phagosome, and NK cell-mediated cytotoxicity) were downregulated by aging, but this was attenuated by MIT-001 treatment (blue cluster). In the green cluster, DEGs downregulated in old mice and upregulated by MIT-001 treatment were associated with the GO terms Transcription, Neurogenesis, and UBl conjugation pathway. In addition, several DEGs (e.g., Grem1, Inhba, and Tgfbr1) in the purple cluster related to TGF-β signaling were moderately upregulated by MIT-001 treatment, similar to those in the green cluster.

Moreover, common DEGs were significantly enriched in the canonical pathways (Reactome pathways [22]), Neutrophil Degradation (Arhgap9, Cd300a, Cd68, Cfd, Clec12a, Dock2, and Fcer1g), Fc-gamma Receptor (FCGR) Dependent Phagocytosis (Fcgr4, Fgr, Hck, Nckap1l, and Pdl4), Immunoregulatory Interactions between a Lymphoid and a Non-Lymphoid Cell (Cd300a, Fcgr4, H2-Q10, Itgb2, Pilra, and Trem2), Interleukin-3, Interleukin-5, and GM-CSF Signaling (Hck, Ptpn6, and Vav1), GPVI-mediated Activation Cascade (Fcer1g, Lcp3, Ptpn6, Rac2, and Vav1), ROS and RNS Production in Phagocytes (Ncf4, Nos3, Rac2, and Slc11a1), DAP12 Signaling (Lcp2, Trem2, and Tyrobp), and Complement Cascade (C1qa, C1ab, C1qc, Cfd, and Vtn). Meanwhile, DEGs enriched in Signaling by ERBB2 (Pik3r1, Prkce, and Yes1) were remarkedly downregulated in the old group and upregulated in the treated group (Figure 2F and Appendix A).

### 2.3. MIT-001 Treatment Alleviates Inflammatory Responses in Old Ovaries

Recently, various transcriptome analyses have shown that aging is associated with a range of immune regulation impairments in human tissues and cells, including the ovaries [23]. Our transcriptome analysis suggested that MIT-001 improved gene expression associated with age-related transcriptome variations in ovaries. We performed gene set enrichment analysis (GSEA) using a set of DEGs (*n* = 1968) identified from two comparisons: old vs. young and treated vs. old. DEGs associated with GO terms linked with immune system regulation were significantly downregulated in the old group compared with the young group. However, these DEGs were upregulated in the treated old group compared with the old group (Figure 3A). Notably, MIT-001 downregulated DEGs enriched in the GO terms Tumor Necrosis Factor Super Family Cytokine Production (*n* = 38; Figure 3B) in old ovaries, including Fcer1g, Slamf9, Clec4a3, Ptpn6, Tyrobp, Adipoq, Bcl3, Trem2, and Sash3. Intriguingly, common DEGs in the two comparisons (old vs. young and treated vs. old) included Lcp2, H2-Ab1, Itgb2, Tyrobp, Cd74, Zap70, Rac2, Cd14, Prpn6, and Fcer1g, which were recently identified as biomarkers of ovarian aging in mice [24]. Moreover, marker genes of old ovaries (Ptpn6, Fcer1g, Tyrobp, Slamf9, and Clec4a3) were downregulated in ovaries of MIT-001-treated old mice, supporting the notion that MIT-001 mitigates transcriptional enhancement of the immune response induced by aging in ovaries (Figure 3C).

In addition, DEGs in aged ovaries, particularly those enriched in GO terms related to ovarian function (Ovarian Follicle Development, Oogenesis, and Ovulation), the T-helper cell-mediated immune response (T-helper 1 Type Immune Response and T-helper 17 Type Immune response), mitochondrial homeostasis (Mitochondrial Calcium Ion Homeostasis and Mitochondrial Membrane Permeability), and aging, were transcriptionally attenuated by MIT-001, albeit only partially with significance (Figure 4).

### 2.4. Cytokine Array Analysis

Cytokines were analyzed in ovarian lysates of young, old, and MIT-001-treated old mice. The membrane protein arrays were incubated with these ovarian lysates (Figure 5A). Cytokine expression differed between ovarian lysates of old and young mice and between those of old and MIT-001-treated old mice. The levels of cytokines were compared between the samples based on the blot intensities. Several cytokines including CD30L, CD30, CRG-2, IGFBP6, IL-1α, IL-1β, IL-2, IL-3, IL-3RB, IL-13, IL-17, KC, leptin R, leptin, L-selectin, Ltn/XCL1, MCP-5, MIP-1α, MIP-1γ, SDF-1α, TARC, TECK, TIMP-1, TNF-α, TPO, TPO, and VCAM-1 were upregulated in ovaries of old mice. However, they were downregulated in the ovaries of MIT-001-treated old mice, often significantly (*p* < 0.05) (Figure 5B). These results show that MIT-001 elicits anti-inflammatory effects in the ovaries of old mice.

## 3. Discussion

In this study, we reported that MIT-001 has an anti-inflammatory effect on the ovaries of old mice and recovers ovarian function. Our results demonstrated that MIT-001 enhanced the development of antral follicles in the ovaries of old mice. Whole transcriptome next-generation sequencing revealed that expression of inflammation-related molecules and inflammatory signaling were decreased in the ovaries of MIT-001-treated old mice.

Old ovaries exhibit several defects such as elevated oxidative stress, mitochondrial dysfunction, fibrosis, inflammation, reduced steroidogenesis, and perturbed follicular development [25,26,27,28]. They contain a reduced number of follicles and poor-quality oocytes, leading to infertility and menopause. A therapeutic tool is needed for aged patients to have a healthy pregnancy and live birth following in vitro fertilization (IVF). Suppression of inflammation in old ovaries may help to restore ovarian function for pregnancy. Several factors, such as programmed cell death and fibrosis, are linked with inflammation in ovaries. Therefore, anti-inflammatory factors are useful for recovering the functions of old ovaries and follicular development. Indeed, the ovaries of MIT-001-treated old mice contained significantly more antral follicles than those of old mice. The number of oocytes per ovary increased two-fold in MIT-001-treated old mice.

We next investigated the mechanism by which MIT-001 enhances follicular development by performing mRNA sequencing. Transcriptomic analysis demonstrated significant transcriptional alterations in the ovaries of old mice (1315 genes) and MIT-001-treated old mice (1197 genes) in comparison with the ovaries of young and old mice, respectively. We determined which GO terms were significantly enriched among these genes. Ptpn6 encodes protein tyrosine phosphatase, nonreceptor type 6, which is identified primarily in hematopoietic and epithelial cells and is overexpressed in ovarian epithelial tissue. Its role in normal ovaries is unclear [24]. PTPN6 is a key regulatory protein in cellular signal transduction in the control of inflammation and cell death. MIT-001 treatment suppressed the expression of Ptpn6 in mouse ovaries. Fcer1g encodes an Fc fragment of IgE receptor Ig, which is associated with innate immunity. It is also involved in cancer immune infiltration and tumorigenesis regulation by the microenvironment. Fcer1g was upregulated in the ovaries of old mice. The role of Fcer1g has not been clearly determined. Our data suggest that Fcer1g is related to inflammation during ovarian aging. MIT-001 downregulated Fcer1g in the ovaries of old mice. Tyrobp encodes TYRO protein tyrosine kinase-binding protein and is an immune-related gene that dysregulates the activation of multiple types of immune cells (T cell, B cells, and natural killer cells) and differentiation of immature cells such as osteoclasts [29]. Tyrobp was dramatically upregulated in ovaries of old mice and downregulated in those of MIT-001-treated old mice. MIT-001 may normalize the immune reaction and thereby recover the function of old ovaries. H2-AB1 was recently reported to be a biomarker of ovarian aging based on the construction of a protein-protein interaction network. H2-Ab1 is part of a hub gene network associated with aging and neurodegenerative disease.

Ovaries exhibit several defects with advanced maternal age, including calcium homeostasis, necrotic cell death, and mitochondrial DNA damage by reactive oxygen species (ROS) [30]. These defects induce ovarian inflammation and impair follicular development, leading to ovarian dysfunction [11]. Dysfunction of ovaries and mitochondria play a major role in these processes. Inflammatory responses can trigger changes in mitochondrial function and dynamics. For example, pro-inflammatory cytokines can induce mitochondrial dysfunction, leading to the production of ROS, which can cause oxidative damage in ovaries. In turn, mitochondrial damage and dysfunction can trigger inflammatory responses. Inflammation and fibrosis are the main causes of dysfunction in old ovaries [11]. Based on the GO data, mitochondrial function was recovered in the ovaries of MIT-001-treated old mice.

During aging, the balance between inflammatory and specific immune responses is disrupted, leading to decreased efficiency of the latter. Cytokines are involved in many aspects of the immune response, including activation and proliferation of immune cells. Therefore, several cytokines are modulated during inflammation and apoptosis in immune cells and tissues. Production and regulation of cytokines are tightly controlled because dysregulation of cytokine production can lead to a variety of immune disorders and diseases. Overproduction of cytokines can lead to inflammation, while underproduction of cytokines can lead to immune suppression and increased susceptibility to infection. Based on the cytokine array data, inflammation occurred in the ovaries of old mice. Several pro-inflammatory and inflammation-related factors are linked with ovarian aging. MIT-001 significantly downregulated cytokines involved in the inflammatory reaction in the ovaries of old mice. CD30 and CD30L are important for innate immune cells involved in inflammation [31]. Various types of deregulated signaling by members of the IL-1 family cause devastating diseases characterized by severe acute or chronic inflammation [32]. TARC and TNF-α were upregulated in the ovaries of old mice compared with those of young mice [33]. TNF-α modulates TIMP-1 during aging. The pro-inflammatory cytokine VCAM-1 is highly expressed upon inflammation and in various chronic conditions together with TNF-α and ROS. Taken together, these results demonstrate that MIT-001 affects many cytokines involved in inflammatory aging via stimulating anti-inflammatory signaling in ovaries. These data help to explain how MIT-001 restores ovarian functionality, such as oogenesis and follicular development, in old mice.

## 4. Materials and Methods

### 4.1. Animals

BDF1 mice (C57BL/6 × DBA/2; F1) were purchased from Orient Bio Co., Ltd. (Seoul, Republic of Korea). All animal experiments, breeding, and care procedures were performed following the regulations of the Institutional Animal Care and Use Committee (IACUC) of CHA University. IACUC approval (approval number IACUC200175) was obtained before the initiation of the study. Young (10 head, 8 weeks old) and old (20 head, 65 weeks old) BDF1 female mice were utilized in this study. Old mice were intraperitoneally injected with MIT-001 (0.5 mg/kg/day) for 1 week (20 head, 65 weeks old).

### 4.2. Superovulation and Collection of GV Oocytes

To induce superovulation, young, old, and MIT-001-treated old female mice were intraperitoneally injected with 7.5 IU PMSG (G4527; Sigma-Aldrich, Saint Louis, MO, USA) at 6 p.m. GV oocytes were retrieved from ovaries at 48 h after PMSG injection. We collected the ovaries of young (6 ovaries), old (11 ovaries), and MIT-001-treated old (11 ovaries).

### 4.3. Histological Analysis

Ovaries of young (*n* = 3), old (*n* = 3), and MIT-001-treated old (*n* = 3) mice were fixed with 4% paraformaldehyde for 30 min at room temperature, dehydrated in an ethanol series (70%, 80%, 95%, and 100% ethanol), embedded in paraffin, and stained with H&E using routine protocols. H&E staining of the tenth and sixteen cross-sections of each sample ovary was performed to count the number of follicles. Images were acquired using an inverted light microscope (Eclipse Ti2; Nikon, Tokyo, Japan) equipped with a camera (DS-Ri2, Nikon) and imaging software (NIS-Elements ver. 4.4., Nikon).

### 4.4. RNA-Seq and Data Analysis

For each group of ovaries (*n* = 2), RNA-seq libraries were prepared using a TruSeq Stranded mRNA LT Sample Prep Kit (Illumina, 20020594, San Diego, CA, USA), as described previously [34]. Total RNA was extracted from ovarian tissue using TRIzol^®^ Reagent (15596026, Thermo Fisher, Waltham, MA, USA) and sequencing libraries were prepared according to the manufacturer’s instructions. Paired-end 101 bp sequencing was performed on an Illumina platform (NovaSeq 6000, Illumina) with duplicate experiments for each group. The raw read data were trimmed using Trimmomatic [35] and the following criteria: first trimming: base quality <3, window size = 4, and mean quality = 15; second trimming: min length = 36 bp. Trimmed reads were aligned to the UCSC mm10 database using HISAT2 [36] and quantified with StringTie [37] using gene/transcript-based annotations from NCBI Annotation Release 108.

Differential expression analysis was performed using DESeq2 [38] with the following parameters: baseMean counts > 14, false discovery rate (FDR) < 0.1, and absolute log2FoldChange > 1. GSEA was conducted using the clusterProfiler packages [39] in R/Bioconductor with gene sets from GO (biological process, cellular component, and molecular function) and canonical pathways (KEGG [40], Reactome [22], and WikiPathway [41]).

### 4.5. Reverse-Transcription qPCR

Ovaries of young (5 ovaries), old (14 ovaries), and MIT-001-treated old (14 ovaries) mice were used for gene expression analysis. Reverse-transcription qPCR was performed to validate the mRNA sequencing results. Based on KEGG pathway analysis results, primers were selected for Ptpn6, Fcer1g, Tyrobp, Slamf2, and Clec4a3 (Appendix A). Total RNA was extracted from ovarian tissue using TRIzol^®^ Reagent (15596026, Thermo Fisher) and reverse-transcribed into cDNA using AccuPower^®^ CycleScript RT PreMix (K-2050, Bioneer, Daejeon, Republic of Korea). qPCR was performed using AccuPower GreenStar qPCR PreMix (K-6212, Bioneer) on a spectrofluorometric thermal cycler (CFX96 Touch Real-Time PCR Detection System, Bio-Rad, Hercules, CA, USA). The reaction contained 10 pmol/µL forward and reverse primers, 200 ng template cDNA, and AccuPower GreenStar qPCR PreMix. Distilled water was added up to a final volume of 20 µL. The PCR cycling conditions were as follows: 3 min at 95 °C, followed by 40 cycles of denaturation for 10 s at 95 °C, and annealing and extension for 20 s at 60 °C. Expression of each gene was normalized to that of β-actin. Triplicate samples were tested. All experiments were repeated three times for statistical analysis with CFX Maestro ver 2.5 (Bio-Rad).

### 4.6. Cytokine Protein Array

Ovaries of young (4 ovaries), old (10 ovaries), and MIT-001-treated old (10 ovaries) mice were homogenized with lysis buffer for the cytokine array experiment. A cytokine array was performed using a Mouse Cytokine Antibody Array Kit (#ab193659; Abcam, Cambridge, MA, USA). The array was used to simultaneously detect 96 mouse cytokines in each homogenized sample according to the manufacturer’s instructions. Ovarian lysates were prepared by homogenization with iNtRON protein extraction buffer (17081; iNtRON Biotechnology, Seongnam, Republic of Korea). Briefly, array membranes were incubated for 30 min at room temperature in 1× blocking buffer (all reagents were supplied with the kit) and then with the tissue lysates for 2 h. The following steps were all performed at room temperature, and all wash stages comprised three washes in wash buffer I for 5 min each, followed by two washes in wash buffer II for 5 min each. The mixture of biotinylated anti-mouse cytokine antibodies was added to the membranes and incubated overnight at 2–8 °C. Membranes were washed and incubated with streptavidin-conjugated horseradish peroxidase overnight at 2–8 °C. Unbound reagents were removed by washing, and membranes were incubated in detection buffer for 2 min. Chemiluminescent signals from bound cytokines were detected by enhanced chemiluminescence. The cytokine signal intensity was measured by spot densitometry using a gel documentation system (WSE-6100 LuminoGraph I; ATTO Tokyo, Tokyo, Japan). Each spot was corrected for adjacent background intensity and normalized to the membrane’s positive control. Each specimen was measured in duplicate, and the mean signal intensity of each cytokine in a specimen was determined. Each blot represents immunoreactive staining with respective antibodies. Staining was absent at the negative control and blank slots. The relative expression level of each cytokine was determined by comparing the pixel intensity of the respective blot to that of the positive control on the same array.

### 4.7. Statistical Analysis

Statistical analysis was performed using a one-way analysis of variance (Holm-Sidak method). The significance level was set at * *p* < 0.05 and ** *p* < 0.005.

## 5. Conclusions

In conclusion, MIT-001 prevents inflammation and enhances follicular development in old ovaries. Despite many studies, an efficient stimulation protocol is lacking for IVF in patients of advanced age. Most treatments that improve pregnancy rates are not recommended in patients of advanced age, and female aging remains one of the most challenging conditions to treat in reproductive medicine. MIT-001 may be applicable to treat age-related female infertility. New controlled ovarian stimulation protocols incorporating MIT-001 should be investigated to enhance follicular development for IVF in aged infertile and prematurely aged female patients.

## Figures and Tables

**Figure 1 ijms-24-15158-f001:**
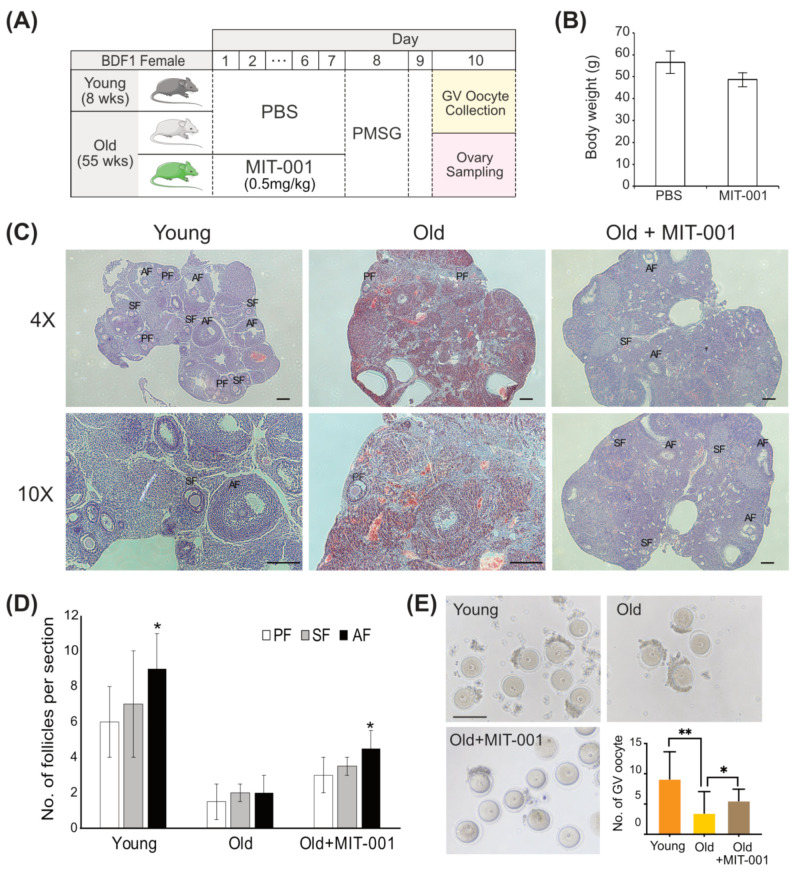
Follicular development in ovaries of MIT-001-treated old mice. (**A**) Schematic overview of MIT-001 treatment of young and old mice. (**B**) Bar graph of the body weights of control and MIT-001-treated old mice. (**C**) H&E-stained ovarian sections (4×) from each group of mice at 48 h after PMSG injection (upper panel) and high magnifications (10×) of mature secondary follicles (bottom panel) in the red boxes by using 3 ovaries from each sample. (**D**) Bar graph of the numbers of primary follicles (PF), secondary follicles (SF), and antral follicles (AF) per ovarian section in each group of mice. (**E**) Brightfield images of GV oocytes collected from each group of mice at 48 h after PMSG injection and bar graph of the number of GV oocytes collected from each group of mice (6 ovaries of young, 11 ovaries of old, and 11 ovaries of MIT-001-treated old mice). Scale bar, 200 µm. * *p* < 0.05 and ** *p* < 0.005.

**Figure 2 ijms-24-15158-f002:**
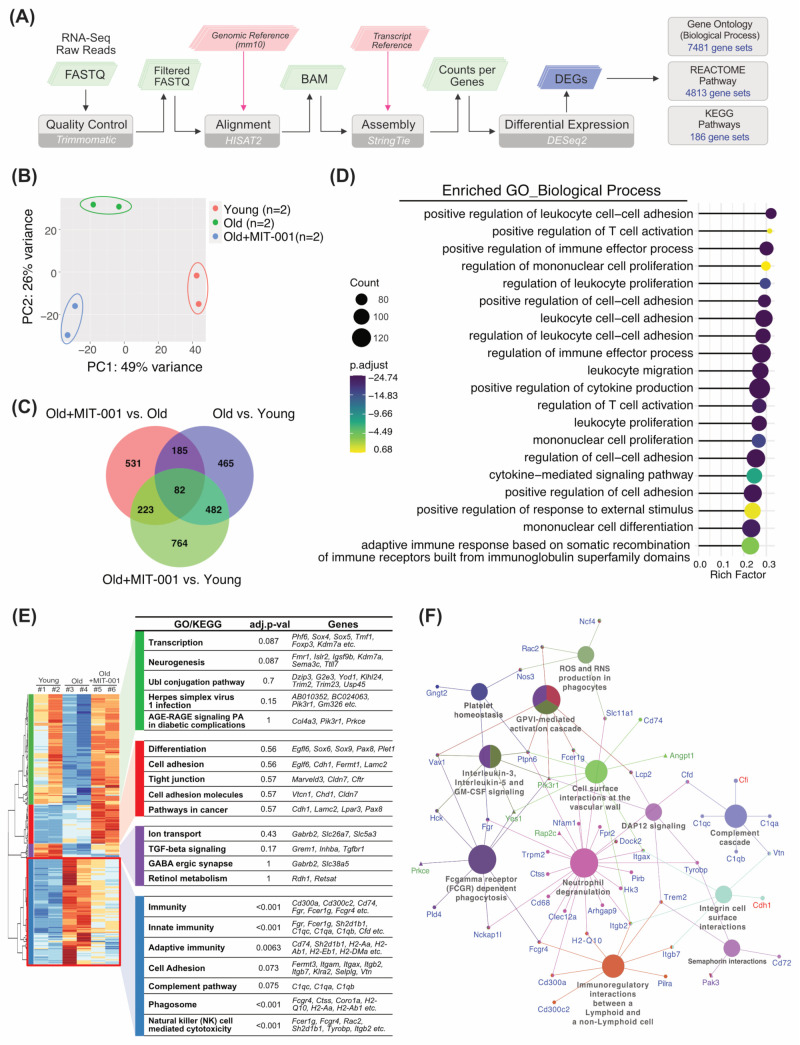
Differential gene expression signatures of ovaries of MIT-001-treated old mice. (**A**) Schematic diagram of RNA-seq and data analysis. (**B**) Scatter plot showing the overall effect of experimental covariates and batch effects by principal component analysis. (**C**) Venn diagram of common or unique DEGs in three comparisons (old vs. young, MIT-001-treated old vs. old, and MIT-001-treated old vs. young) (log2 fold change > 2, FDR < 0.05). (**D**) Lollipop chart showing the rich factors of enriched biological process GO terms for the overall DEGs (*n* = 2732). (**E**) Clustered heatmap of common DEGs between two comparisons (old vs. young and MIT-001-treated old vs. old) (*n* = 1968). Enriched GO/KEGG terms and representative DEGs in each colored cluster (green, red, purple, blue) are displayed in the table. (**F**) Interactive plot displaying enriched Reactome pathways for the clustered DEGs featured in the heatmap of (**E**). Gene symbols connected to enriched pathway terms are colored corresponding to the cluster groups.

**Figure 3 ijms-24-15158-f003:**
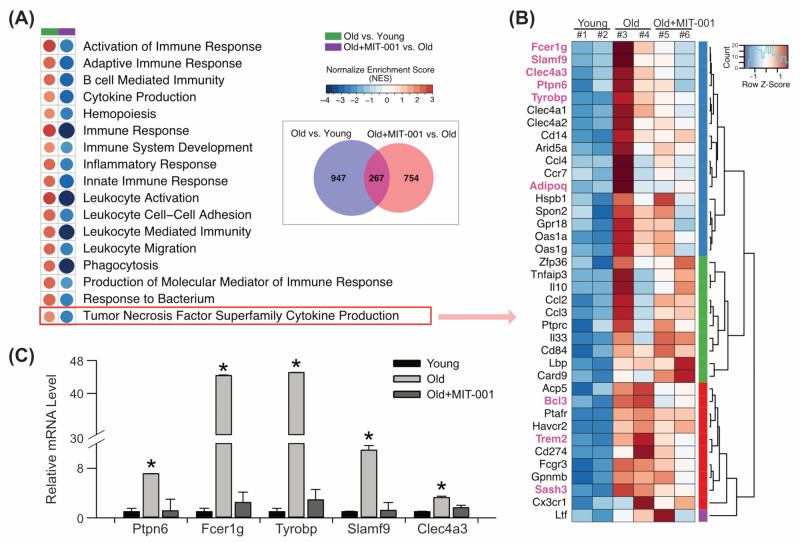
Effect of MIT-001 on TNF superfamily cytokine signaling in ovaries of old mice. (**A**) Circle heatmap of enriched immune-associated GO terms for DEGs in each comparison (old vs. young and treated vs. old). The colors denote the log2 fold changes of DEGs. (**B**) Clustered heatmap of DEGs enriched in the GO term Tumor Necrosis Factor Superfamily Cytokine Production that highlighted by the red box in (**A**). Pink genes correspond to common DEGs in both comparisons. (**C**) Bar graph of mRNA expression of ovarian aging marker genes. * *p* < 0.05.

**Figure 4 ijms-24-15158-f004:**
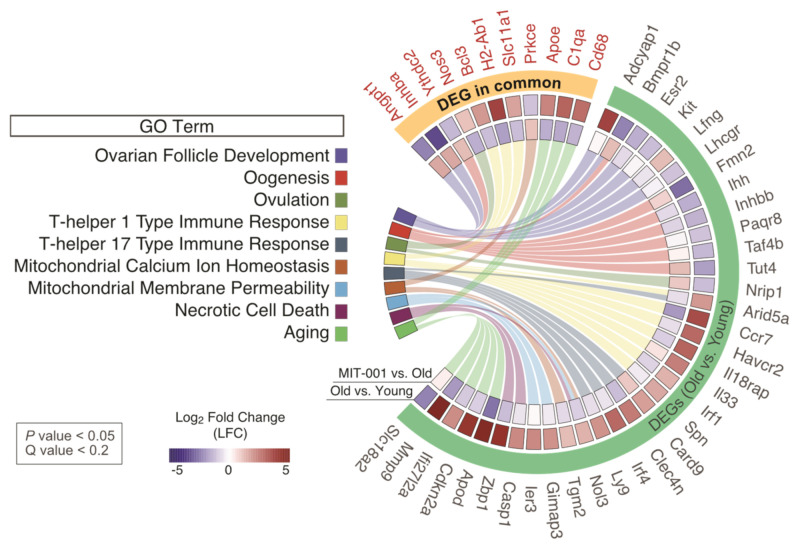
Chord diagram of gene expression related to ovarian function and aging obtained by GO enrichment analysis. Chord diagram represents connections between GO terms and enriched DEGs associated with ovarian function, MIT-001-specific immune responses, and aging. In the outer arc, DEGs are separated into two clusters (orange and green). One cluster includes DEGs in both comparisons (old vs. young and treated vs. old) and the other cluster includes DEGs in only one comparison (old vs. young). In the two inner arcs, colors represent fold changes in each comparison.

**Figure 5 ijms-24-15158-f005:**
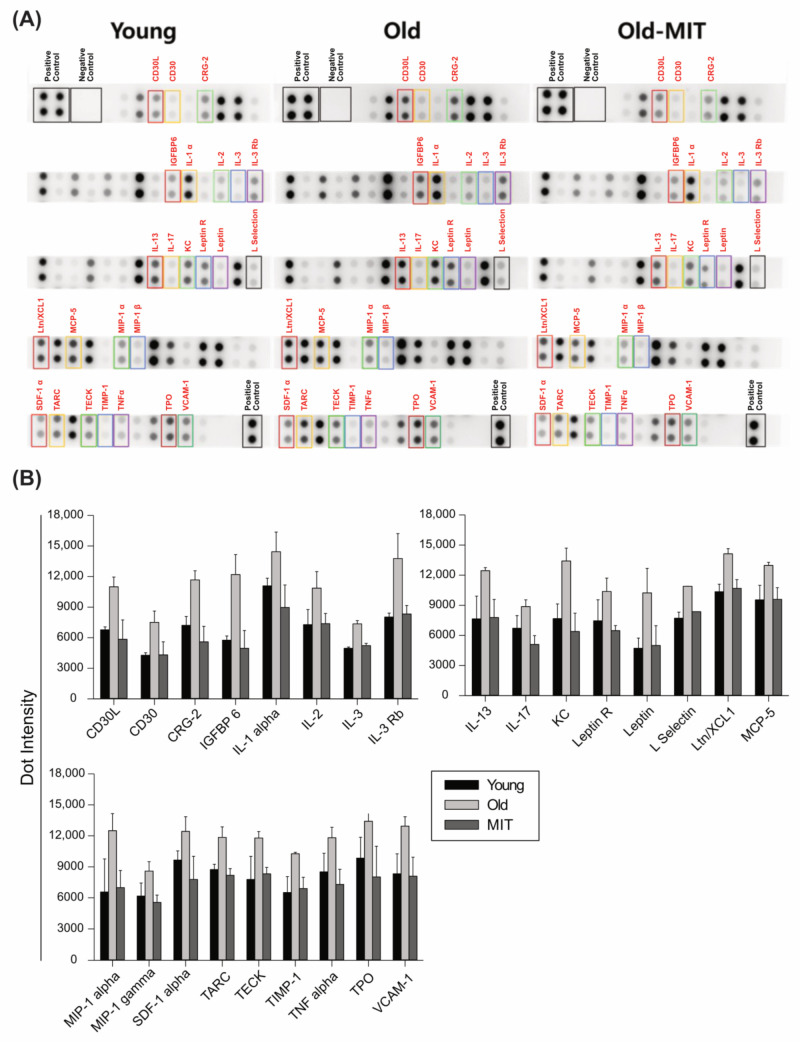
Representative images of cytokine array blots (**A**) Assessment of 96 cytokines using a Mouse Cytokine Array Kit. Representative images of cytokine array blots are shown. MIT-001 significantly downregulated several cytokines (indicated by colored squares), including CD30L, CD30, CRG-2, IFGBP6, ILs, KC, leptin R, L-selectin, Ltn/XCL1, MCP-5, MIP-5, MIP-α, MIP-1γ, SDF-1α, TARC, TECK, TIMP-1, TNF-α, TPO, and VCAM-1, in ovaries of old mice. The positive and negative controls confirmed the reliability of the data. Data from four repeats were used for statistical analysis. (**B**) The bar graph presents the quantification of the dot intensities in the corresponding blots depending on the samples. The results in young and MIT-001-treated old mice were significantly different (*p* < 0.05) from those in old mice.

## Data Availability

The datasets analyzed during the current study are available in the GEO repository (https://www.ncbi.nlm.nih.gov/geo/query/acc.cgi?acc=GSE232918, accessed on 12 October 2023).

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
