# Peer review of "Mitigating Age-Related Ovarian Dysfunction with the Anti-Inflammatory Agent MIT-001"

_ijms, 2023, doi:10.3390/ijms242015158_

Round 1
Reviewer 1 Report
In the current manuscript, the authors studied the effect of the MIT-001 drug on old and young mice for the development of follicles and oocytes. The authors extracted the RNA and proteins from the oocytes and studied the expression levels of various genes. The manuscript is written well; however, it needs thorough revision. I have listed a few points that may help to improve the manuscript.
1. The study design needs significant improvement. The authors collected Oocytes from young (n=8), old (n=20), and MIT-001-treated old (n=20) mice ovaries but it is not clear why they used only n=3 from each study group for downstream study. In the PCA plot, there are only two subjects were shown.
2. The authors used young, old, and old+MIT-001-treated mice to study the follicular development in ovaries. Figure 1C does not really demonstrate there is a significant improvement in SF, PF, and AF compared to the old subjects. It is also unclear why the H&E has different levels of staining in three groups when they are done simultaneously.
3. How were the differential expression of the genes in the three study groups determined and what criteria were used? The authors could demonstrate this by showing simple volcano plots. Also, why do the old and old+MIT001 show significant variance in the PCA plot?
4. Figure 2C needs more discussion, Why the authors are comparing DEGs across the three groups, and what is the significance of this Venn diagram? The authors may also want to explain what rich factor matrices are and how confident it is.
5. The authors may want to revise the conclusion.
6. The authors extracted proteins and used them for cytokine array assay, however, it would be more valuable if some of those samples could be used for proteomic analysis to verify the real translation of RNA seq data.
Author Response
- The study design needs significant improvement. The authors collected Oocytes from young (n=8), old (n=20), and MIT-001-treated old (n=20) mice ovaries but it is not clear why they used only n=3 from each study group for downstream study. In the PCA plot, there are only two subjects were shown.
We would like to express our gratitude to the reviewer for their valuable comments. We sincerely apologize for the oversight in not providing an explanation regarding the number of ovary samples used in each downstream analysis, which may have caused confusion. As a response to the reviewer's valuable feedback, we have supplemented the 'Materials and Methods' section by including the number of ovaries used for each analysis. Additionally, for RNA sequencing analysis, two ovaries from each group were utilized, resulting in the representation of two subjects per group in the PCA plot.
- The authors used young, old, and old+MIT-001-treated mice to study the follicular development in ovaries. Figure 1C does not really demonstrate there is a significant improvement in SF, PF, and AF compared to the old subjects. It is also unclear why the H&E has different levels of staining in three groups when they are done simultaneously.
We appreciate the valuable comments from the reviewer. We have updated the new HE pictures in Figure 1C to provide a more effective demonstration of follicle development, specifically regarding the follicle development ratios for each condition.
- How were the differential expressions of the genes in the three study groups determined and what criteria were used? The authors could demonstrate this by showing simple volcano plots. Also, why do the old and old+MIT001 show significant variance in the PCA plot?
Thanks to the reviewer for valuable comments to improve overall quality of the manuscript and data figures. First, we described the criteria in DESeq2 tool for the differential expression of the genes in the three study groups in the materials and methods section as follows: baseMean counts > 14, false discovery rate (FDR) < 0.1, and absolute log2FoldChange > 1. Second, as the comments from the reviewer, we could have demonstrated differentially expressed genes using volcano plots for comparisons between each group. However, we made efforts to provide a more comprehensive view by quantifying the number of genes that changed commonly or specifically in comparisons between each group. Finally, as we conducted principal component analysis (PCA) on all differentially expressed genes significantly altered through group comparisons, we represented the results using PCA plots. Consequently, there is a significant difference in the expression profiles of differentially expressed genes between the old group and the old group treated with MIT-001.
- Figure 2C needs more discussion, Why the authors are comparing DEGs across the three groups, and what is the significance of this Venn diagram? The authors may also want to explain what rich factor matrices are and how confident it is.
ïƒ Once again thanks to the reviewer for encouraging comments about our work and presentation of results.
Response to your first comment, we aimed to emphasize that out of the differentially expressed genes altered through group comparisons, only 82 genes were consistently changing with respect to the aging level of mice and drug treatment, as revealed in the Venn diagram. Also, we have removed the red-line section indicated within the venn diagram as it could potentially lead to misunderstanding in the interpretation of subsequent result plots.
Response to your second comment, we used rich factor in scatter plot of GO terms associated with biological process, which is the ratio of differentially expressed gene numbers annotated in this GO term to all gene numbers annotated in this GO term. The greater the rich factor, the greater the degree of GO terms enrichment.
In addition, we represented the plots based on enrichment test results that met the criteria of p.adjust < 0.05 and q value < 0.2
- The authors may want to revise the conclusion.
We would like to express our gratitude to the reviewer for their suggestions aimed at enhancing the overall appeal of the manuscript for readers. We have incorporated the reviewer's comments into the manuscript. However, I did not find any significant differences that would warrant a revision of the conclusion from my perspective.
- The authors extracted proteins and used them for cytokine array assay, however, it would be more valuable if some of those samples could be used for proteomic analysis to verify the real translation of RNA seq data.
We extend our gratitude to the reviewer for their excellent comments. Our study primarily centers around immune-related factors, notably cytokines. This choice is guided by the comprehensive RNA sequencing results, which conspicuously reveals several immune response factors associated with aging and the rejuvenation of aging phenotypes through the administration of anti-inflammatory agents. It's important to note that various cytokines are secreted from organs into the bloodstream. As a result, the mRNA sequencing results provide only limited information on cytokine expression. To gain a more comprehensive understanding of the functional immune factors, such as cytokines, with respect to immune modulation, we intend to investigate their protein levels.
Once again thanks to the reviewer for wonderful comments to improve the overall quality of the manuscript.
Reviewer 2 Report
The abstract and introduction is well written.
Please revise the last paragraph of the introduction. It seem to contain comments from a previous reviewer:"The introduction should briefly place the study in a broad context and highlight why it is important. It should define the purpose of the work and its significance. The current state of the research field should be carefully reviewed and key publications cited. Please highlight controversial and diverging hypotheses when necessary. Finally, briefly mention the main aim of the work and highlight the principal conclusions. As far as possible, please keep the introduction comprehensible to scientists outside your particular field of research. References should be numbered in order of appearance and indicated by a numeral or numerals in square brackets—e.g., [1] or [2,3], or [4–6]. See the end of the document for further details on references."
Discrepancy in the age of old mice in method section and Figure 1a. Please correct it. Also, it is unclear how many mice were used in the total study and at what age "Young (n=20, 8 weeks old) and old (n=40, 65 weeks old) BDF1 female 323 mice were utilized in this study. Old mice were intraperitoneally injected with MIT-001 324 (0.5 mg/kg/day) for 1 week. Then we collected the ovaries of young (n=20), old (n=40), and 325 MIT-001-treated old (n=40) mice." Is the 20+40 or 20+40+40? Please mention the "n" in figure 1a. Also, mention number of mice used for PMSG in 1a.
Were the old mice aged 55 weeks or 65 weeks? different parts of the manuscript has different information.
Typo in line 337 - Sixtieth ?
Please explain the importance of different follicles with respect to reproduction as your agent MIT-001 is claimed to restore the function of old ovaries and improve female fertility. In my opinion the figure 1c and 1d have a disconnect with regards to the no of follicles shown in the images and graphed. Please choose better representative images and please be clear in the comparison for the statistics in 1d.
Figure 2: Please provide justification for using 2 samples from each group.
Figure 5: P value missing in the graph.
Author Response
The abstract and introduction is well written.
Please revise the last paragraph of the introduction. It seem to contain comments from a previous reviewer: "The introduction should briefly place the study in a broad context and highlight why it is important. It should define the purpose of the work and its significance. The current state of the research field should be carefully reviewed and key publications cited. Please highlight controversial and diverging hypotheses when necessary. Finally, briefly mention the main aim of the work and highlight the principal conclusions. As far as possible, please keep the introduction comprehensible to scientists outside your particular field of research. References should be numbered in order of appearance and indicated by a numeral or numerals in square brackets—e.g., [1] or [2,3], or [4–6]. See the end of the document for further details on references."
We apologize for the inclusion of an unnecessary sentence in the introduction. We have since deleted the sentence.
Discrepancy in the age of old mice in method section and Figure 1a. Please correct it. Also, it is unclear how many mice were used in the total study and at what age "Young (n=20, 8 weeks old) and old (n=40, 65 weeks old) BDF1 female 323 mice were utilized in this study. Old mice were intraperitoneally injected with MIT-001 324 (0.5 mg/kg/day) for 1 week. Then we collected the ovaries of young (n=20), old (n=40), and 325 MIT-001-treated old (n=40) mice." Is the 20+40 or 20+40+40? Please mention the "n" in figure 1a. Also, mention number of mice used for PMSG in 1a.
We express our utmost gratitude to the reviewer for their meticulous review of the manuscript. We deeply regret any inaccuracies in our previous statements. We have taken note of the concern raised regarding the number of subjects used in our study. At present, we are in the process of revising the manuscript to provide a more transparent explanation regarding the number of mice and ovaries involved in our experiments. To clarify, 'n' represents the number of ovaries. Specifically, we utilized 10 young mice, 20 old mice, and 20 old mice treated with MIT-001. Consequently, we collected 20 ovaries from the young mice and 40 ovaries each from the old mice and the MIT-001 treated old mice. We will update the manuscript, including relevant figures, to present this information more clearly.
Were the old mice aged 55 weeks or 65 weeks? different parts of the manuscript has different information.
We apologize for any confusion earlier. The correct timeframe is indeed 65 weeks. We have updated the manuscript accordingly to reflect this accurate information.
Typo in line 337 - Sixtieth?
We appreciate the reviewer for pointing out the mistake. We revised the incorrect writing sixteen.
Please explain the importance of different follicles with respect to reproduction as your agent MIT-001 is claimed to restore the function of old ovaries and improve female fertility. In my opinion the figure 1c and 1d have a disconnect with regards to the no of follicles shown in the images and graphed. Please choose better representative images and please be clear in the comparison for the statistics in 1d.
Thank you for your comment. We have updated the new HE pictures to provide a clearer demonstration of the follicle development ratios for each condition.
Figure 2: Please provide justification for using 2 samples from each group.
Figure 5: P value missing in the graph.
We sincerely appreciate the reviewer's meticulous examination of each plot. We have revised the figures to accurately depict the number of samples, and we have also updated the p-values in the graphs.
Once again thanks to the reviewer for wonderful comments to improve the overall quality of the manuscript.
Round 2
Reviewer 1 Report
The revised version is much improved and can be accepted.